# Ultra-Fine Polyethylene Hernia Meshes Improve Biocompatibility and Reduce Intraperitoneal Adhesions in IPOM Position in Animal Models

**DOI:** 10.3390/biomedicines10061294

**Published:** 2022-05-31

**Authors:** Marius J. Helmedag, Daniel Heise, Roman M. Eickhoff, Sophia M. Schmitz, Mare Mechelinck, Caroline Emonts, Tim Bolle, Thomas Gries, Ulf Peter Neumann, Christian Daniel Klink, Andreas Lambertz

**Affiliations:** 1Department of General, Visceral and Transplantation Surgery, RWTH Aachen University Hospital, 52074 Aachen, Germany; dheise@ukaachen.de (D.H.); reickhoff@ukaachen.de (R.M.E.); sopschmitz@ukaachen.de (S.M.S.); uneumann@ukaachen.de (U.P.N.); christian.klink@diakonissen.de (C.D.K.); alambertz@ukaachen.de (A.L.); 2Department of Anesthesiology, Uniklinik RWTH Aachen, 52074 Aachen, Germany; mmechelinck@ukaachen.de; 3Institut für Textiltechnik, RWTH Aachen University, 52074 Aachen, Germany; caroline.emonts@ita.rwth-aachen.de (C.E.); tim.bolle@ita.rwth-aachen.de (T.B.); thomas.gries@ita.rwth-aachen.de (T.G.)

**Keywords:** polyethylene mesh, biocompatibility, IPOM, ultrafine diameter filaments, adhesion formation, animal trials

## Abstract

(1) Introduction: The intraperitoneal onlay mesh technique (IPOM) is widely used to repair incisional hernias. This method has advantages but suffers from complications due to intraperitoneal adhesion formation between the mesh and intestine. An ideal mesh minimizes adhesions and shows good biocompatibility. To address this, newly developed multifilamentous polyethylene (PET) meshes were constructed from sub-macrophage-sized monofilaments and studied regarding biocompatibility and adhesion formation. (2) Methods: We investigated fine (FPET, 72 filaments, 11 µm diameter each) and ultra-fine multifilament (UFPET, 700 filaments, 3 µm diameter each) polyethylene meshes for biocompatibility in subcutaneous implantation in rats. Adhesion formation was analyzed in the IPOM position in rabbits. Geometrically identical mono-filamentous polypropylene (PP) Bard Soft^®^ PP meshes were used for comparison. Histologic and immune-histologic foreign body reactions were assessed in 48 rats after 7 or 21 days (four mesh types, with two different mesh types per rat; *n* = 6 per mesh type). Additionally, two different mesh types each were placed in the IPOM position in 24 rabbits to compile the Diamond peritoneal adhesion score after the same timeframes. The biocompatibility and adhesion score differences were analyzed with the Kruskal–Wallis nonparametric statistical test. (3) Results: Overall, FPET and, especially, UFPET showed significantly smaller foreign body granulomas compared to PP meshes. Longer observation periods enhanced the differences. Immunohistology showed no significant differences in the cellular immune response and proliferation. UFPET demonstrated significantly reduced peritoneal adhesion formation compared to all other tested meshes after 21 days. (4) Conclusions: Overall, FPET and, especially, UFPET demonstrated their suitability for IPOM hernia meshes in animal models by improving major aspects of the foreign body reaction and reducing adhesion formation.

## 1. Introduction

Surgical meshes are irreplaceable devices in the treatment of hernia. According to expert consensus, all fascia defects regardless of etiology larger than 2 cm should be augmented by mesh reinforcement [1]. The recurrence rate of incisional hernia (IH), especially, benefits from the introduction of meshes to support the abdominal wall after surgical treatment [2]. The current standards of practice are large-pore, monofilament, lightweight synthetic meshes, especially for extraperitoneal applications [3]. The disadvantages of placing synthetic meshes as a foreign material into the abdominal wall include mesh-related complications such as mesh infection, mesh shrinkage due to excessive scar formation, and chronic pain [4]. Focusing on intraperitoneal placement, as in the intraperitoneal onlay mesh technique (IPOM), bowel adhesions and fistula formation are highly relevant complications [5,6]. Numerous meshes are recommended for intraperitoneal applications, but the optimal mesh for intraperitoneal use has still not been found [7]. In order to avoid the aforementioned complications, various composite meshes are marketed to reduce bowel adhesions in the IPOM position [3,8]. Usually, a polymer such as polypropylene (PP)—which is known to induce adhesion formation and to promote tissue ingrowth—is combined with a second layer to be placed toward the visceral side in order to reduce tissue adhesions. For example, a resorbable layer of sodium hyaluronate, carboxy methylcellulose, and polyethylene glycol hydrogel is added to prevent visceral adhesions in Sepramesh™. Other absorbable coatings designed for this application include oxidized cellulose for Proceed^®^ and collagen for Parietex™ meshes [9]. Different approaches consist of either permanent coatings such as titanium TI-Mesh or the employment of a low-adhesion inductive polymer such as polyvinylidene fluoride (PVDF) towards the visceral side [10]. It has additionally been shown that the extent of intraperitoneal adhesion formation is determined not only by the material in contact with the viscera but also by pore size and surface area [11,12]. Although some multifilaments tend to cause an increased foreign body reaction, recent intriguing research by Eickhoff et al. revealed that non-circular shaped “snowflake” sutures or multifilaments with ultrafine individual polyethylene terephthalate (PET) strands could attenuate the foreign body reaction and lead to smaller foreign body granulomas (FGBs) [13]. It is hypothesized that sub-macrophage-sized surface patterns lead to these desirable outcomes. Therefore, new PET meshes were specifically developed with ultrafine monofilament diameters to possibly employ the advantages of this mesh material and configuration. In our research, we first aimed to investigate the biocompatibility of multifilament PET meshes in “fine” and “ultrafine” configurations by subcutaneous implantation in rats compared to two different standard meshes. Secondly, we examined the effect of these multifilament meshes on adhesion formation in the IPOM position in a laparoscopic rabbit model. Two different monofilament polypropylene (PP) meshes were identically investigated for comparison.

## 2. Materials and Methods

### 2.1. Experimental Specimens

Two newly manufactured multifilament PET meshes in so-called fine (FPET) and ultra-fine (UFPET) configurations were investigated in our experiments. Identical multifilament diameters of 90 dtex were used. The filament count (FPET: 72; UFPET: 700) and diameter (FPET: 11 µm; UFPET: 3 µm) varied accordingly (see Table 1). Mesh geometry, e.g., porosity, pore size, and mesh surface, matched the configurations of previously clinically approved employed standard meshes. The PET meshes were compared to mono-filament PP meshes with an identical warp knitting pattern and commercially available Bard^®^ Soft Meshes (Davol Inc., Warwick, RI, USA 02886). The meshes were produced on an RS4 net raschel machine (KARL MAYER Holding GmbH & Co. KG, Obertshausen, Germany) with a machine gauge of E14. After production, the mesh samples were characterized regarding pore size, porosity, and thickness. A sample size of five was applied for the following methods. The pore size and porosity were determined with light microscopy (Leica M205C, Leica Camera AG, Wetzlar, Germany). The pictures were converted to gray-scale pictures and analyzed using the Leica Application Suite software, V 3.8. The thickness was determined according to DIN EN ISO 5084.

### 2.2. Animal Experiments

All animal trials were conducted according to German legislation regarding animal studies and were approved by the state governments Animal Care and Use Committee (see Institutional Board Review Statement).

### 2.3. Surgical Procedures

#### 2.3.1. Rat Surgeries

In total, 48 female Sprague–Dawley rats (200–300 g) were held under standardized conditions conforming to EU directive 2010/63/EU ABD ETS 123: a temperature between 22 and 24 °C, a relative humidity of 50–60%, and a 12-h light/12-h darkness cycle. The rats had access to water and food ad libitum. Two different mesh types were implanted per rat for each observational period of 7 or 21 days. Each mesh type was tested six times for each timeframe (*n* = 6). All the procedures were carried out under general anesthesia, which was applied in a transparent induction box via the inhalation of an isoflurane (2 to 3% by volume) oxygen mixture (97% by volume). Then, the weight was determined. The rats were positioned on a temperature-controlled heating pad in the supine position, and an isoflurane (1 to 2 % by volume) oxygen (97%) air mixture device was administered via nosecone for continued narcosis. For intraoperative and postoperative analgesia, 100 mg/kg bodyweight of metamizole was subcutaneously injected. Appropriate anesthesia depth was checked by testing the reflexes of the paw and eyelid. In order to protect the eyes and avoid corneal dehydration, a dexpanthenol ointment (Bepanthen ^®^ Roche eye and nose ointment, Hoffman-La Roche, Grenzach-Wyhlen, Germany) was applied in the conjunctival sac to the animals, and the eyelids were closed. The belly region was shaved and disinfected by the application of an antiseptic. (Octenisept; Schülke, Norderstedt, Germany). Two different mesh specimens of about 2 cm^2^ were subcutaneously placed after midline skin incision and the lateral preparation of subcutaneous pockets. Meshes were kept at a distance of 2 cm to ensure that the specimens were not cross-influenced. The surgeon supervised rats until complete recovery from the procedure. The animals were sacrificed after 7 or 21 days of observation, and the abdominal wall was resected and prepared for histological evaluation.

#### 2.3.2. Rabbit Surgeries

Overall, 24 female New Zealand White rabbits were included in our study. Two different mesh types were implanted in each rabbit. Meshes were tested in sextuplicate (*n* = 6) for each observational period of 7 or 21 days. The average bodyweight was around 3000 g at time of surgery. The animals were housed in rooms with a constant temperature and a controlled day/night cycle. They received the usual rabbit feed, as well as water and hay ad libitum. The animals were kept in cages under normal laboratory conditions, with regular checks by the staff of the experiment and the supervising veterinarian. The socialization of the animals was carried out whenever possible. Furthermore, the animals were kept at least in pairs. Postoperatively, all animals were visited daily. Laparoscopic IPOM placement was performed as previously described [14]. In brief, anesthesia was applied via a subcutaneous injection of a combination of 0.1 mL (0.1 mg) Domitor (Farmos, Turku, Finland) per kg bodyweight at 1 mg/mL and 0.2 mL (20 mg) per kg bodyweight of 10% Ketamin (Ketamin^®^ 10%, Fa. SanofiCeva, Düsseldorf, Germany). Then, 0.2 mL (0.01 mg)/kg/h of fentanyl was continuously infused via an intravenous access. After intubation, volume-controlled ventilation with an isoflurane (2% by volume)—oxygen (50% by volume)—air mixture was performed with a ventilator. The rabbits were fixated in the supine position on a temperature-controlled heating pad. The abdomen was shaved and disinfected with a polyvidone-iodine solution. A 10-mm optical trocar was placed by mini laparotomy in the midline of the epigastrium, and the pneumoperitoneum was established (CO_2_ insufflator Richard Wolf, Knittlingen, Germany). Two additional trocars were introduced in the right and left upper quadrant. (Figure 1) First, a peritoneal area of around 1.5 by 0.5 cm was incised and removed in both lower quadrants to induce adhesion formation. These defects were laparoscopically covered by a random different combination of meshes. All meshes were fixated by laparoscopic tacks (AbsorbaTack^®^, 5 mm, Covidien, Mansfield, TX, USA) (Figure 2). The abdominal cavity was checked for any bleeding or injury, and the trocars were removed. The abdominal wall and skin were closed by suturing. After the completion of the respective experimental period of 7 or 21 days, all rabbits were euthanized using a lethal dose of 400 mg/kg bodyweight (2.5 mL/kg bodyweight) of pentobarbital (Narcoren^®^, 16 g pentobarbital sodium/100 mL, Rhône Merieux, Laupheim, Germany). A midline laparotomy was performed, and the adhesion score was recorded by two observers blinded to the study group. Abdominal walls with meshes were explanted and fixated for possible future investigations.

### 2.4. Histology and Immunohistochemistry

Histological preparation and standard and immunohistochemical staining for hematoxylin and eosin, macrophage markers (CD68), and proliferation (KI-67) were performed as previously described [13]. In short, after 48 h of 4% formaldehyde fixation, 3-μm paraffin slides were prepared and stained. The immune response to implanted filaments was further characterized by the staining of the T-cell receptor and lymphocyte markers (CD 45+). We used anti-mouse rabbit secondary antibodies (1:300; Dako ^®^, Glostrup, Denmark) for all immunohistochemical stainings. The size of the FBG was calculated as stated by Eickhoff et al. [13]. Each FBG was measured twice in each specimen. All microscopic imaging was undertaken with the TissueFAXS Plus upright fluorescence and brightfield system (TissueGnostics GmbH., Vienna, Austria). The percentage of positive cells was determined with a software-supported, semi-automatic fashion (Strataquest, TissueGnostics GmbH) for immunohistochemical stainings by first marking the regions of interest (ROI) of the same size around the implanted sutures. Cells were identified by detecting the dark blue nuclei in the cell associated with the morphological detection of the cell membranes in the ROI. Cells were counted as marker-positive if each slide had a certain specific threshold eclipsed by the immunohistochemical dye; hence, a percentage of positive cells for each immunochemical staining and each ROI could be measured.

### 2.5. Adhesion Score

The postoperative peritoneal adhesions were scored as previously established by Diamond et al. [15]. Three observers independently scored the adhesions, and majority scores were used in the final analysis. Scoring categories consisted of the extent, type, and tenacity of the adhesions and were added for a total Diamond score. For exact scoring, please refer to Table 2.

### 2.6. Statistical Analysis

Statistical comparisons between the experimental groups for the mesh characterization were performed as one-way ANOVAs with post-hoc LSD tests with IBM SPSS Statistics 25 (IBM Corp., Armonk, NY, USA). Statistical comparisons for the biocompatibility assessment and Diamond scores were performed as non-parametric one-way ANOVAs (Kruskal–Wallis) against the null hypothesis that there were no differences between the groups. Post-hoc testing was carried out using Dunn’s multiple-comparison testing method. GraphPad 8 (GraphPad Software Inc., San Diego, CA, USA) was used for these statistical analyses. Significance thresholds for all comparisons were set at 0.05.

## 3. Results

### 3.1. Mesh Characterization

The morphological characteristics of the mesh were highly dependent on the used fiber material and textile properties. Using multifilament yarns with varying filament counts and diameters significantly influenced morphological characteristics, such as thickness, pore size, and porosity (Figure 3).

The thickness of the meshes made from multifilament yarns, PET (0.335 ± 0.003 mm) and UFPET (0.308 ± 0.015 mm), were significantly lower than the thickness of the PP monofilament mesh (0.597 ± 0.021 mm, *p* < 0,001), as well as the commercially available Bard^®^ Soft Mesh (0.453 ± 0.015 mm, *p* < 0.001).

Increasing the filament count of the meshes led to a significantly higher porosity compared to the monofilament meshes. The porosity of the UFPET mesh (7.44 ± 1.12%) was significantly higher than each monofilament mesh (Bard^®^ Soft Mesh: 66.26 ± 0.62%; PP: 66.57 ± 1.07%, *p* < 0.001) and the PET mesh (68.92 ± 1.39%, *p* < 0.001).

The pore size of the meshes was significantly increased by using multifilament yarns. A further increase in the pore size was shown for finer single fiber diameter in the multifilament yarn. The pore size of the UFPET meshes (3.88 ± 0.48 mm^2^) was significantly larger than those of the PET mesh (3.33 ± 0.14 mm^2^, *p* = 0.013) and the monofilament meshes (PP: 2.91 ± 0.33 mm^2^; Bard^®^ Soft Mesh: 2.76 ± 0.13 mm^2^, *p* < 0.001). There were no significant differences in the pore size and porosity of the monofilament meshes.

### 3.2. Biocompatibility Testing in Rats

Subcutaneous mesh implantation surgeries did not suffer from complications or infections for the observational periods of 7 or 21 days. All animals and specimens could be included in the analysis.

#### 3.2.1. Foreign Body Granuloma

The inner foreign body granuloma (IFBG) is the histologic representative of the cellular infiltrate around foreign materials. The outer foreign body granuloma (OFBG) is the resulting collagen-rich scar tissue largely produced by the cells of the IFBG. A significant reduction in IFBG size was recorded for UFPET (36.1 ± 8.0 µm) compared to the Bard Soft Mesh (53.4 ± 18.0 µm, *p* < 0.05) and the PP mesh (60.7 ± 18.7 µm, *p* < 0.01) after 7 days. The accompanying OFBG was significantly smaller for FPET (108.6 ± 32.0 µm) than PP (163.9 ± 35.1 µm, *p* < 0.05) and UFPET (99.0 ± 48.3 µm) than PP (*p* < 0.01). Additionally, the IFBG of UFPET (33.3 ± 14.25 µm) had a significantly reduced size compared to PP (52.1 ± 12.4 µm, *p* < 0.01) after 21 days. The OFBG measurements after 21 days furthermore revealed significantly smaller measurements for UFPET (86.0 ± 40.2 µm) than FPET (152.4 ± 40.2 µm, *p* < 0.01), UFPET than Bard Soft (167.1 ± 47.6 µm, *p* < 0.001), and UFPET than PP (170.8 ± 25.8 µm, *p* < 0.001). All other comparisons showed no statistically significant differences for 7 and 21 days. For clarity and comprehensibility, please refer to Table 3 and Figure 4.

#### 3.2.2. Immunohistology Analysis

The cellular reaction surrounding the implanted meshes was characterized by immunohistology markers identifying various cells of the immune system and general proliferation activity. The percentage of positive cells in the respective ROIs did not significantly differ for macrophage (CD 68+), T-cell (CD 3+), lymphocyte (CD 45+), or proliferative activity (KI 67+) after 7 or 21 days. Please refer to Figure 5 and Figure 6 for comparisons and. additionally, to Table 4 for the descriptive statistics.

### 3.3. Adhesion Assessment in Rabbits

All rabbits quickly recovered from the surgical intervention. One rabbit had to be euthanized on postoperative day 2 due to complications not related to the meshes (strangulation of intestine by herniation through a laparoscopic incision). The rabbit was replaced by an additional animal to keep the experimental groups at equal size. The FPET and UFPET meshes were easily pliable and provided exceptional handling properties, especially compared to the stiffer PP mesh. The blinded assessment of the Diamond score revealed no significant differences between extent, type, tenacity, or total score after 7 days. In contrast, the extent of adhesion was significantly reduced for UFPET (median: 0.5) compared to PP (median: 1.5) after 21 days (*p* < 0.05). The tenacity score of adhesion was significantly lower for UFPET (median: 0.5) than PP (median: 3, *p* < 0.05) and for UFPET than Bard Soft (median: 3, *p* < 0.05) after 21 days of elapsed implantation time. This resulted in a significantly lowered total Diamond score for UFPET (median: 2) than PP (median 6.5, *p* < 0.01) and significantly lower total Diamond score for UFPET than Bard Soft (median: 7, *p* < 0.05). Please refer to Figure 7 and Figure 8 for a comprehensive scatter plot of the individual results.

## 4. Discussion

Adhesion formation between viscera and implanted meshes and succeeding complications are still a major drawback of IPOM hernia repair [5,6]. The amount and properties of resulting adhesions are determined by a complex interaction between the material and textile characteristics of the meshes [11,12]. The research of Eickhoff et al. demonstrated how textile modifications of suture strands—in particular, the surface structures of sub-macrophage-sized patterns—led to an improved foreign body reaction [13]. The enlarged surface area of polymer filaments consisting of filaments larger than macrophages induces and increases foreign body reactions, even for biomaterials that are considered to be of good biocompatibility, such as PP [16]. Our results further illuminate this ostensible contradiction: UFPET meshes led to decreased scar induction in biocompatibility assessments, as indicated by the reduced IFBG and OFBG after 7 days and the smaller IFBG after 21 days but especially evidenced by the significant reduction in the OFBG compared to all other tested meshes after 21 days. Since the dreaded adhesion formation is closely related to granuloma induction, it is apt that the intraperitoneal adhesions formed in contact with the UFPET were also significantly smaller than PP or Bard Soft Mesh after 21 days, as demonstrated by the significant lowering of the Diamond adhesion score. In comparison, we could not confirm this significant effect on adhesion score after the shorter timeframe of 7 days due to the continuing scar adhesion formation and maturation until up to at least 3 weeks. Taken together, the presented results support the hypothesis of Eickhoff et al. [13] that surface patterns, such as microfibers smaller than macrophages (7 µm), impede the cellular response toward foreign materials, even though the interface area for possible reactions is absolutely increased. More specifically, we illustrated how a filament consisting of microfibers of around 3 µm in diameter can ameliorate foreign body response in a multifilament mesh constructed according to these principles. An identical mesh constructed out of slightly larger-sized filaments of around 11 µm (FPET) did not lead to identical improvements in adhesion formation and did not show similar decreased granuloma formation in biocompatibility experiments. It can be trivially argued that an enlarged cross-section of an implanted biomaterial fiber usually leads to an increased foreign body response [17,18] simply due to the fact that this usually means larger quantities of biomaterial for the host to react to. Just minimizing the amount of biomaterial used is not easily feasible, because mechanical constraints and desired stability for hernia meshes require certain minimal yarn diameters for mesh construction that are dependent on warp knitting and mechanical properties due to the polymer and the multi- or monofilament nature of these yarns. These differences result in diverging yarn diameters between the PP and PET meshes, although the linear yarn weight is similar. The results, especially in the foreign body granuloma, cannot be solely explained by the different diameters and polymers alone, as there are also significant differences between FPET and UFPET meshes, although their diameter and polymer are essentially the same. Adhesiogenesis is a complex process mediated by cellular and cytokines that leads to the mesothelial-to-mesenchymal transformation (MMT) of the mesothelial cells of the peritoneum [19]. This transformation is the endpoint of a pathological reaction towards peritoneal trauma or inflammation. Milky spots in the omentum, as zones of macrophage and lymphocyte aggregation, are additional recruitment areas for additional cells relevant in the formation of lasting adhesions [20]. It is still unclear how different mesh parameters influence these biological players in the adhesiogenesis cascade, especially because we did not find significant differences in the cell characterizations in our histology. However, our findings underscore how surface pattering in the micrometer range can be an alternative area of design optimization beyond just reducing the volume of foreign material that may be similar to material or porosity adjustments [3].

Our study was limited concerning longer time frames, which are certainly relevant, because mesh implantation is supposed to permanently stay once put into place. Nonetheless, we believe significant deductions can be drawn from this study, because as far as we know, there has been no research that indicates a reversal of behavior regarding adhesion formation. Additionally, the foreign body reaction and the timeframes employed in this study are well-established for mesh assessment in the literature [14,21]. Another limitation of our study was a lack of comparisons of microfiber meshes versus meshes specifically designed for intraperitoneal placement with absorbable coatings (such as collagen or oxidized cellulose) or permanent coatings (such as titanium). In the future, these investigations could further illuminate whether a similar adhesion avoidance could be generated by UFPET meshes in relation to more complex and possibly expensive multi-component meshes.

## 5. Conclusions

The quest for the “ideal” mesh for hernia surgery is a complex task, as many design parameters such as material, porosity, filament size, and count and surface area interact to create the foreign body response towards a mesh [3]. Our results demonstrate that a new hernia mesh employing filaments smaller than macrophages could be constructed and lead to reduced adhesion in a rabbit IPOM model over 21 days. The histologic foreign body response was additionally reduced after 7 and 21 days. This research illuminates the potential of employing fibers of such sizes in order to positively influence mesh properties in an IPOM setting. Further research should further compare these findings to other mesh materials more specifically tailored to IPOM applications, such as titanium-coated meshes.

## Figures and Tables

**Figure 1 biomedicines-10-01294-f001:**
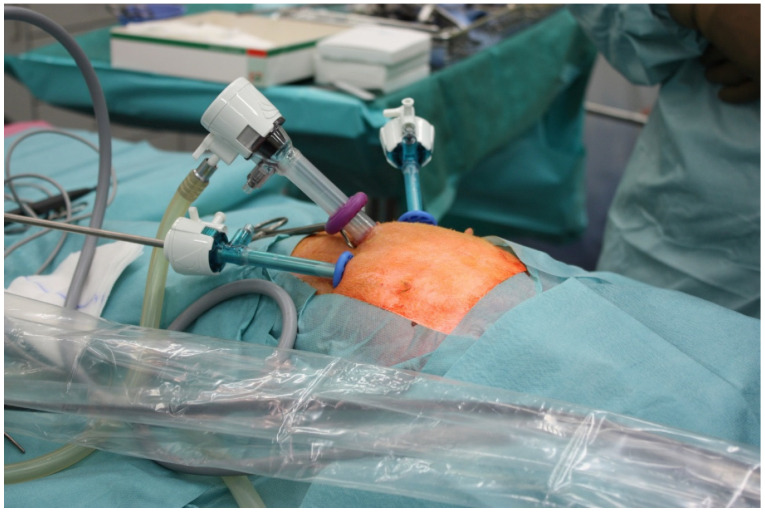
Laparoscopic rabbit suture set-up.

**Figure 2 biomedicines-10-01294-f002:**
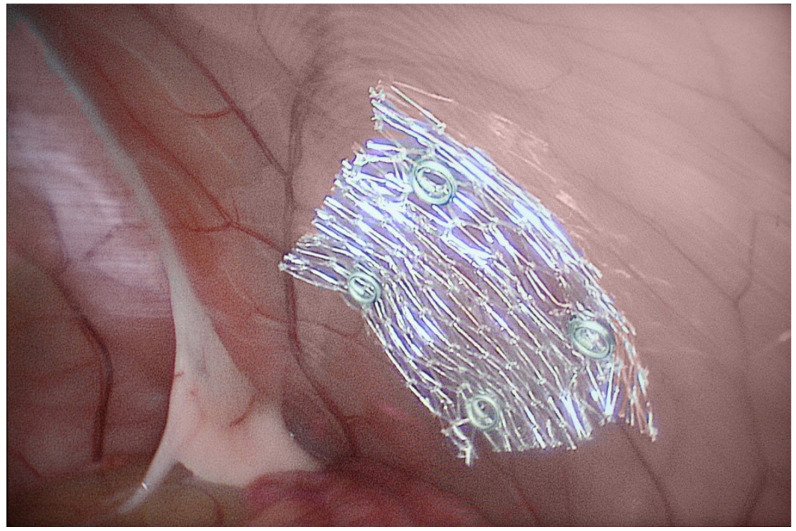
Intraperitoneal placement of ultra-fine PET mesh fixated by surgical tacks.

**Figure 3 biomedicines-10-01294-f003:**
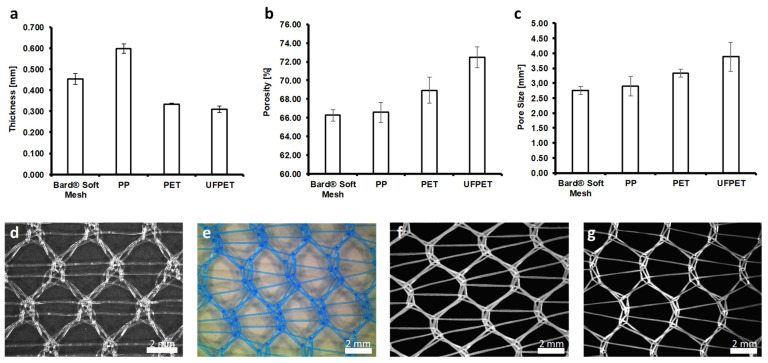
Morphological characterization and light microscopic images of the different mesh configurations: (**a**) thickness, (**b**) porosity, (**c**) pore size, (**d**) Bard^®^ Soft Mesh, (**e**) PP monofilament mesh, (**f**) PET multifilament mesh, and (**g**) UFPET multifilament mesh.

**Figure 4 biomedicines-10-01294-f004:**
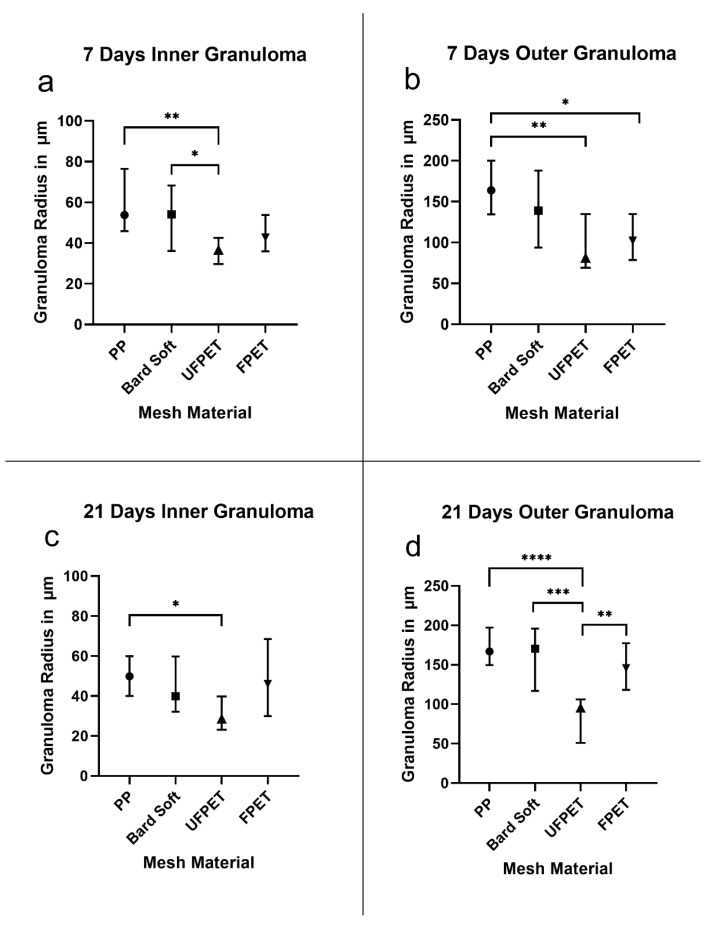
Foreign body granuloma sizes measured in µm. Asterisks denote significance levels of the Kruskal–Wallis analysis. (**a**) 7 days inner granuloma, (**b**) seven days outer granuloma, (**c**) 21 days inner granuloma, (**d**) 21 days outer Granuloma: * *p* < 0.05, ** *p* < 0.01, *** *p* < 0.001, and **** *p* < 0.0001.

**Figure 5 biomedicines-10-01294-f005:**
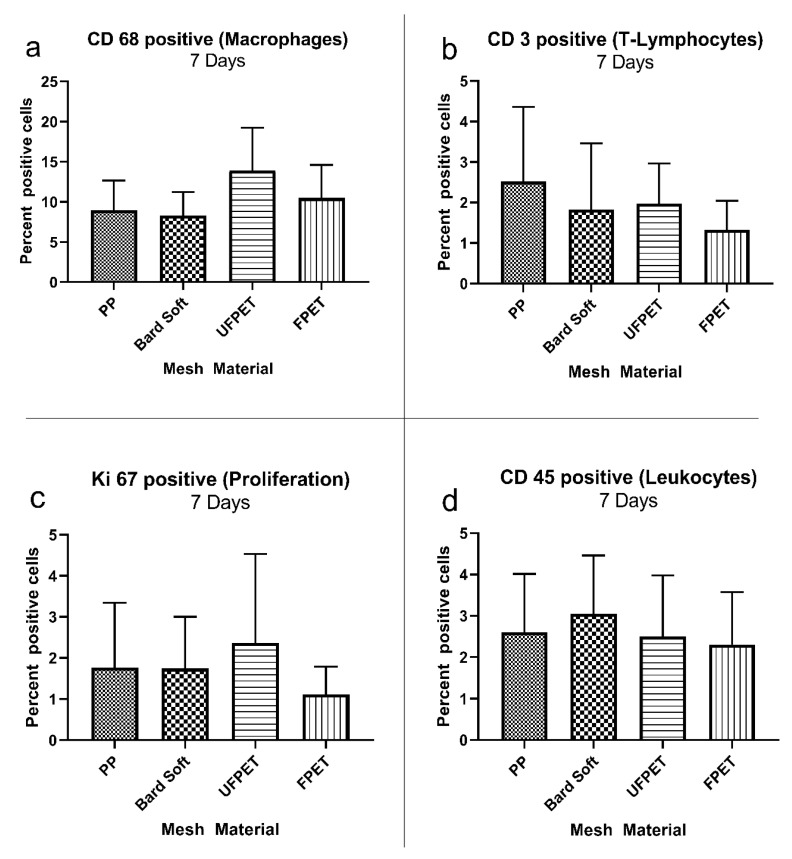
Immuno-histologic characterization of foreign body response after 7 days showing marker-positive cells in the region of interest surrounding the subcutaneously implanted meshes. Kruskal–Wallis comparisons did not reach the significance threshold of *p* < 0.05. (**a**) CD 68 positive (macrophages), (**b**) CD 3 positive (T-lymphocytes), (**c**) KI 67 positive (proliferation), (**d**) CD 45 positive (leukocytes).

**Figure 6 biomedicines-10-01294-f006:**
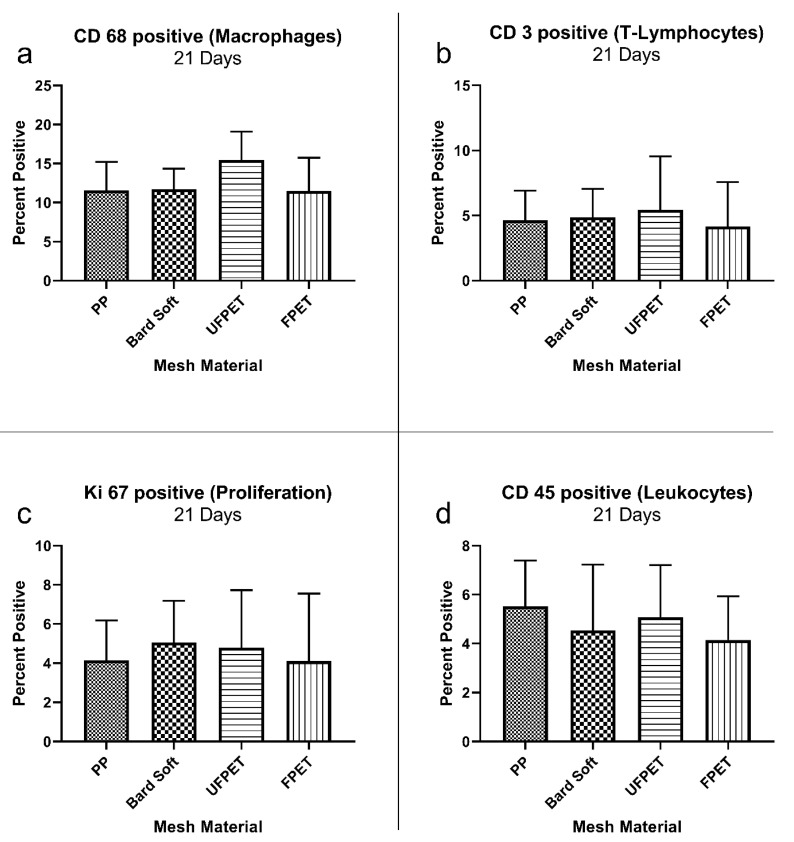
Immuno-histologic characterization of foreign body response after 21 days showing marker-positive cells in the region of interest surrounding the subcutaneously implanted meshes. Kruskal–Wallis comparisons did not reach the significance threshold of *p* < 0.05. (**a**) CD 68 positive (macrophages), (**b**) CD 3 positive (T-lymphocytes), (**c**) KI 67 positive (proliferation), (**d**) CD 45 positive (leukocytes).

**Figure 7 biomedicines-10-01294-f007:**
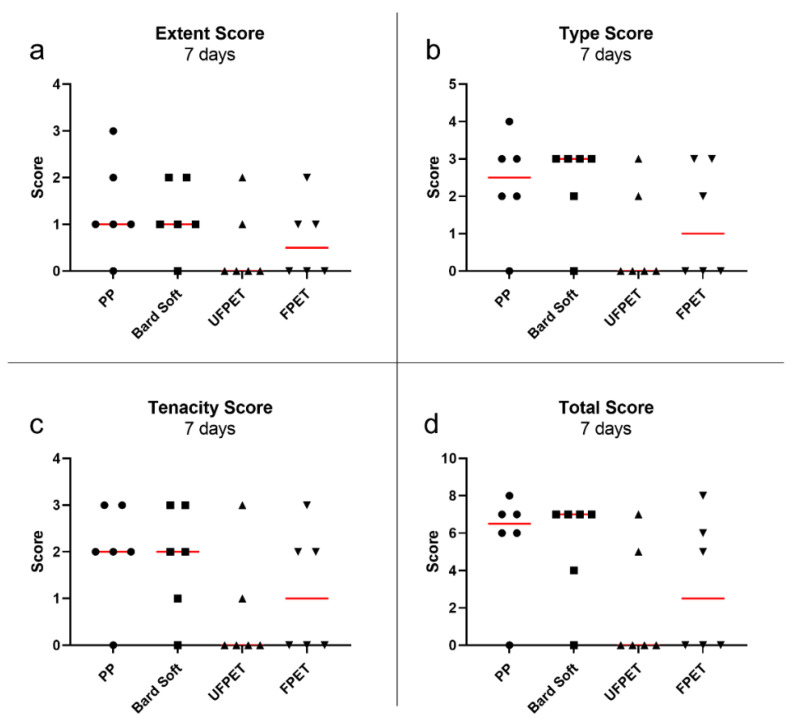
Scatter plot of Diamond adhesion scores after 7 days. Red bar denotes median. Kruskal–Wallis comparisons did not reach the significance threshold of *p* < 0.05. (**a**) Extent score, (**b**) Type score, (**c**) Tenacity score, (**d**) Total score.

**Figure 8 biomedicines-10-01294-f008:**
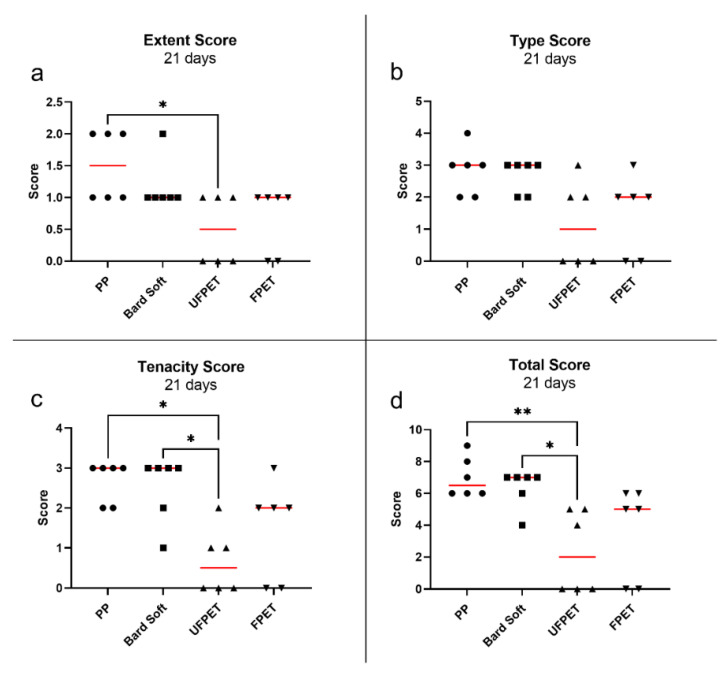
Scatter plot of Diamond adhesion scores after 7 days. Red bar denotes median. * *p* < 0.05 and ** *p* < 0.01. (**a**) Extent score, (**b**) Type score, (**c**) Tenacity score, (**d**) Total score.

**Table 1 biomedicines-10-01294-t001:** Fiber characteristics of various meshes.

Mesh	Bard^®^ Soft Mesh	Ultra-Fine PET	Fine PET	PP Monofilament
Material	Polypropylene	PET	PET	Polypropylene
Filament size (dtex)	105	90	90	108
Filament count	1	700	72	1
Ø single filament (µm)	121	3	11	122

**Table 2 biomedicines-10-01294-t002:** Peritoneal adhesion score created according to Diamond et al. [15].

Parameter	Score Points
Extent of site involvement	
None	0
<25%	1
<50%	2
<75%	3
<100%	4
Type	
None	0
Filmy, transparent, avascular	1
Opaque, translucent, avascular	2
Opaque, capillaries present	3
Opaque, larger vessels present	4
Tenacity	
None	0
Adhesion falls apart	1
Adhesion lysed with traction	2
Adhesion requiring sharp dissection	3
Possible total	11

**Table 3 biomedicines-10-01294-t003:** Average inner and outer granuloma sizes with the standard deviation, as measured with hematoxylin/eosin staining.

Mesh Type	PP	Bard Soft	FPET	UFPET
After 7 days				
IFBG (µm)	60.7 ± 18.7	53.4 ± 18.0	45.0 ± 9.8	36.1 ± 8.0
OFBG (µm)	163.9 ± 35.1	142.7 ± 42.2	108.6 ± 32.0	99.0 ± 48.3
After 21 days				
IFBG (µm)	52.1 ± 12.4	44.9 ± 18.0	45.0 ± 9.8	33.3 ± 14.25
OFBG (µm)	170.8 ± 25.8	167.1 ± 47.6	152.4 ± 40.2	86.0 ± 40.2

Abbreviations: PP: polypropylene, FPET: fine polyethylene, UFPET: ultra-fine polyethylene, IFBG: inner foreign body granuloma, and OFBG: outer foreign body granuloma.

**Table 4 biomedicines-10-01294-t004:** Descriptive statistics of the immuno-histologic characterization of foreign body responses after 7 and 21 days. Kruskal–Wallis comparisons did not reach the significance threshold of *p* < 0.05.

Mesh Type	PP	Bard Soft	FPET	UFPET
7 days				
CD 68+	8.93 ± 3.72%	8.76 ± 3.39%	10.5 ± 4.13%	13.2 ± 5.5%
CD 3+	2.52 ± 1.84%	1.83 ± 1.63%	1.97 ± 0.99%	1.33 ± 0.715%
KI 67+	1.77 ± 1.58%	1.75 ± 1.25%	1.12 ± 0.68%	2.37 ± 2.16%
CD 45+	2.6 ± 1.42%	3.05 ± 1.41%	2.31 ± 1.27%	2.5 ± 1.48%
21 days				
CD 68+	11.9 ± 3.55%	11.7 ± 2.65%	14.7 ± 3.24%	11.5 ± 4.25%
CD 3+	4.65 ± 2.27%	4.88 ± 2.18%	5.44 ± 4.1%	5.44 ± 4.1%
KI 67+	4.16 ± 2.02%	5.05 ± 2.21%	4.1 ± 3.45%	4.79 ± 2.93%
CD 45+	5.52 ± 1.88%	4.54 ± 2.69%	5.07 ± 2.1%	5.07 ± 2.14%

## Data Availability

Not applicable.

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
