# Peer review of "Ultra-Fine Polyethylene Hernia Meshes Improve Biocompatibility and Reduce Intraperitoneal Adhesions in IPOM Position in Animal Models"

_biomedicines, 2022, doi:10.3390/biomedicines10061294_

Round 1

Reviewer 1 Report

The manuscript titled “Ultra-fine polyethylene hernia meshes improve biocompatibility and reduce intraperitoneal adhesions in IPOM position in animal models” addressed the study on the effect of polyethylene hernia meshes on biocompatibility and intraperitoneal adhesions. My comments on it are as follows:

  1. This current manuscript is dull to read, with many syntax errors.
  2. The authors didn’t addressed the reason PET was used in the introduction section.
  3. Why did the same images for meshes appear in Figure 4 and Figure 9?
  4. The authors mentioned “Enlarged surface area of polymer filaments consisting of filaments larger than macrophages induce and increased foreign body reaction” on page 13 in the discussion section, is it also true for those materials with good biocompatible?
  5. Histological images for adhesion should be also provided for the evaluation.
  6. The right way to compare the effect of surface patterns on adhesion is to use the same surface area of polymer filaments, otherwise, it will lead to wrong conclusions.

Based on the above comments and problems, the current manuscript is not suitable for publication. A careful revision for manuscript will be suggested.

Reviewer 2 Report

In the manuscript, Helmedag et al. reported a kind of Ultrafine polyethylene hernia meshes with improved biocompatibility and reduced intraperitoneal adhesions in IPOM position in several animal models. The authors provided a lot of data regarding the foreign body response and adhesion formation of several hernia meshes. This is a suitable manuscript for publication in Biomedicines. However, extended language editing is necessary before its acceptance.

There are too many spelling errors to numerate throughout the manuscript, especially in the Materials and Methods. The following are some examples.

Page 3 line 131, 0.1 mL (0.1 mL).

Page 5 lin2 162, anti-mouse rabbit

Page 7 line 225, 108.6 μM ± 32.0 μM.

Only histogram bar images were provided for histology and immunohistochemistry. Please include the original data in the manuscript.

Statistical analysis should be performed among groups in Figures 6-10.
